biomathematics/health and disease and epidemiology/theoretical biology

mathematical epidemiology, COVID-19, lockdown, information, human behaviour

# Effects of information-induced behavioural changes during the COVID-19 lockdowns: the case of Italy

## Bruno Buonomo[1] and Rossella Della Marca[2]

[1]Department of Mathematics and Applications, University of Naples Federico II, via Cintia, 80126 Naples, Italy
[2]Department of Mathematical, Physical and Computer Sciences, University of Parma, Parco Area delle Scienze 53/A, 43124 Parma, Italy

BB, 0000-0003-4998-3725; RDM, 0000-0002-8661-0429

The COVID-19 pandemic that started in China in December 2019 has not only threatened world public health, but severely impacted almost every facet of life, including behavioural and psychological aspects. In this paper, we focus on the 'human element' and propose a mathematical model to investigate the effects on the COVID-19 epidemic of social behavioural changes in response to lockdowns. We consider an SEIR-like epidemic model where the contact and quarantine rates depend on the available information and rumours about the disease status in the community. The model is applied to the case of the COVID-19 epidemic in Italy. We consider the period that stretches between 24 February 2020, when the first bulletin by the Italian Civil Protection was reported and 18 May 2020, when the lockdown restrictions were mostly removed. The role played by the information-related parameters is determined by evaluating how they affect suitable outbreak-severity indicators. We estimate that citizen compliance with mitigation measures played a decisive role in curbing the epidemic curve by preventing a duplication of deaths and about 46% more infections.

## 1. Introduction

In December 2019, the Municipal Health Commission of Wuhan, China, reported to the World Health Organization a cluster of viral pneumonia of unknown aetiology in Wuhan City, Hubei province. On 9 January 2020, the China CDC reported that the respiratory disease, later named COVID-19, was caused by the novel coronavirus SARS-CoV-2 [1]. The outbreak of COVID-19 rapidly expanded from Hubei province to the rest of China and then to other countries. Finally, it developed in a devastating

pandemic affecting almost all the countries of the world [2]. As of 14 August 2020, a total of more than 21 million cases of COVID-19 and 764 741 related deaths were reported worldwide [2].

In the absence of a treatment or a vaccine, the mitigation strategies enforced by many countries during the COVID-19 pandemic were based on social distancing. Each government enacted a series of restrictions affecting billions of people, including recommendation of restricted movements for some or all of their citizens, and localized or national lockdown with the partial or full closing-off of non-essential companies and manufacturing plants [3].

Italy was the first European country affected by COVID-19. It was strongly hit by the epidemic, which triggered progressively stricter restrictions aimed at minimizing the spread of the coronavirus. The actions enacted by the Italian government began with reducing social interactions through quarantine and isolation and culminated in a *full lockdown* [4,5]. On 4 May 2020, the *phase two* began, marking a gradual reopening of the economy and restriction easing for residents.

During the period that stretches between 22 January and 14 August 2020, Italy suffered 252 809 official COVID-19 cases and 35 234 deaths [6].

The scientific community promptly reacted to the COVID-19 pandemic. Since the early stage of the pandemic a number of mathematical models and methods was used. Among the main concerns were: predicting the evolution of the COVID-19 pandemic worldwide or in specific countries [7–9]; predicting epidemic peaks and ICU accesses [10]; assessing the effects of containment measures [7–9,11] and, more generally, assessing the impact on populations in terms of economics, societal needs, employment, healthcare, death toll etc. [12,13].

Among the mathematical approaches used, many authors relied on deterministic compartmental models. This approach was successful for reproducing epidemic curves in the past SARS-CoV outbreak in 2002–2003 [14]. Specific studies were focused on the case of the COVID-19 epidemic in Italy: Gatto *et al.* [11] studied the transmission between a network of 107 Italian provinces by using an SEPIA model as a core model. Their SEPIA model discriminates between infectious individuals depending on presence and severity of their symptoms. They examined the effects of the intervention measures in terms of the number of averted cases and hospitalizations in the period 22 February–25 March 2020. Giordano *et al.* [9] proposed a very detailed model, named SIDARTHE, in which the distinction between diagnosed and non-diagnosed individuals plays an important role. They predicted the course of the epidemic and showed the need to use testing and contact tracing combined with social distancing measures.

The mitigation measures for COVID-19 like social distancing, quarantine and self-isolation were encouraged or mandated [8]. Although the vast majority of people were following the rules, even in this last case there were many reports of people breaching restrictions [15,16]. Local authorities needed to continuously verify compliance with mitigation measures through monitoring by health officials and police actions (checkpoints, use of drones, fine or jail threats etc.). This behaviour might be related to costs that individuals affected by epidemic control measures paid in terms of health, including loss of social relationships, psychological pressure, increasing stress and health hazards resulting in a substantial damage to population well-being [12,17].

Modelling the interplay between human behaviour and the spread of infectious diseases is a topic of increasing interest [18,19] and includes recent models focusing on COVID-19. For example, Acuña-Zegarra *et al.* [20] assumed that sanitary emergency measures are implemented at a given time, after which the population splits in two distinct subpopulations depending on whether they adhere or do not adhere to the measures. Inspired by the behavioural economic model by Chen *et al.* [21], an SEIR model was proposed by Suwanprasert [22] where individuals are allowed to optimally choose their public avoidance actions in response to COVID-19 risk.

In this paper, the change in social behaviour is described by employing the method of information-dependent models [23,24] which is based on the introduction of a suitable *information index* (see [23,24]). This method has been applied to vaccine-preventable childhood diseases [24,25] and is increasingly being used (see [26,27] for very recent contributions).

The main goal here is to assess the effects on the COVID-19 epidemic of human behavioural changes during the lockdowns. To this aim, we build an information-dependent SEIR-like model which is based on the key assumption that the choice to respect the lockdown restrictions, specifically the social distance and the quarantine, is partially determined on a fully voluntary basis and depends on the available rumours and information concerning the spread of the COVID-19 disease in the community.

A second goal of this manuscript is to provide an application of the information index to a specific field-case, where the model is parametrized and the solutions are compared with official data.

We focus on the case of the COVID-19 epidemic in Italy during the period that begins on 24 February 2020, when the first bulletin by the Italian Civil Protection was reported [6], includes the partial and full

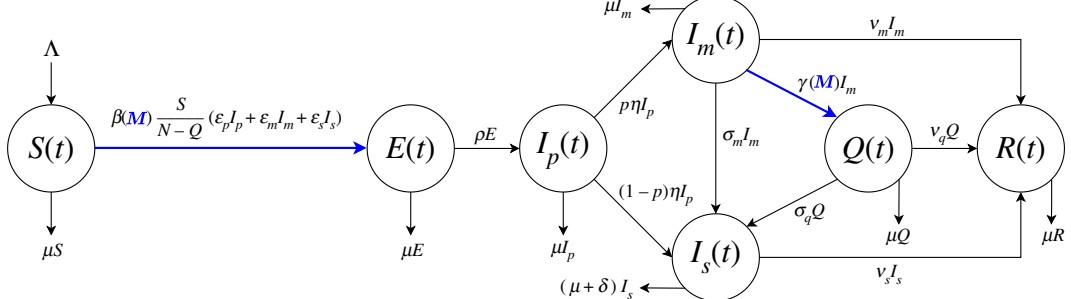

**Figure 1.** Flow chart for the COVID-19 model (2.1)–(2.2). The population $N(t)$ is divided into seven disjoint compartments of individuals: susceptible $S(t)$, exposed $E(t)$, post-latent $I_p(t)$, asymptomatic/mildly symptomatic $I_m(t)$, severely symptomatic (hospitalized) $I_s(t)$, quarantined $Q(t)$ and recovered $R(t)$. Blue colour indicates the information-dependent processes in model (see (2.4)–(2.6), with $M(t)$ ruled by (2.2)).

lockdown restrictions, and ends on 18 May 2020, when the lockdown restrictions were mostly removed. We stress the role played by circulating information by evaluating the absolute and relative variations of disease-severity indicators as functions of the information-related parameters.

# 2. Model formulation

## 2.1. State variables and balance equations

We assume that the total population $N$ is divided into seven disjoint *compartments*: susceptible $S$, exposed $E$, post-latent $I_p$, asymptomatic/mildly symptomatic $I_m$, severely symptomatic (hospitalized) $I_s$, quarantined $Q$ and recovered $R$ (therefore $N = S + E + I_p + I_m + I_s + Q + R$).

The size of each compartment at time $t$ represents a *state variable* of a mathematical model. The state variables and the processes included in the model are illustrated in the flow chart in figure 1. In tables 1 and 2, we provide a description for each parameter. The model is given by the following system of nonlinear ordinary differential equations, where each *balance equation* rules the rate of change of a state variable.

$$\dot{S} = \Lambda - \beta(M)\frac{S}{N-Q}(\varepsilon_p I_p + \varepsilon_m I_m + \varepsilon_s I_s) - \mu S, \tag{2.1a}$$

$$\dot{E} = \beta(M)\frac{S}{N-Q}(\varepsilon_p I_p + \varepsilon_m I_m + \varepsilon_s I_s) - \rho E - \mu E, \tag{2.1b}$$

$$\dot{I}_p = \rho E - \eta I_p - \mu I_p, \tag{2.1c}$$

$$\dot{I}_m = p\eta I_p - \gamma(M)I_m - \sigma_m I_m - \nu_m I_m - \mu I_m, \tag{2.1d}$$

$$\dot{I}_s = (1-p)\eta I_p + \sigma_m I_m + \sigma_q Q - \nu_s I_s - \delta I_s - \mu I_s, \tag{2.1e}$$

$$\dot{Q} = \gamma(M)I_m - \sigma_q Q - \nu_q Q - \mu Q, \tag{2.1f}$$

$$\dot{R} = \nu_m I_m + \nu_s I_s + \nu_q Q - \mu R. \tag{2.1g}$$

The model formulation is described in detail in the next subsections.

## 2.2. The role of information

We assume that the final choice to adhere or not to adhere to lockdown restrictions is partially determined on a fully voluntary basis and depends on the available information concerning the spread of the disease in the community.

The information is mathematically represented by an *information index* $M(t)$ (see electronic supplementary material, §S1 for the general definition), which summarizes the information about the current and past values of the disease [25,26,28] and is given by the following distributed delay:

$$M(t) = \int_{-\infty}^{t} k(Q(\tau) + I_s(\tau))Erl_{1,a}(t - \tau)\, d\tau.$$

This formulation may be interpreted as follows: the first-order Erlang distribution $Erl_{1,a}(x)$ represents an exponentially fading memory, where the parameter $a$ is the inverse of the average time delay $T_a$ of the

**Table 1.** Temporal horizon, initial conditions and epidemiological parameters values for model (2.1)–(2.2).

| parameter | description | baseline value |
|---|---|---|
| $t_0$ | initial simulation time | 24 February 2020 |
| $t_f$ | final simulation time | 18 May 2020 |
| $S(t_0)$ | initial number of susceptible individuals | $60.357 \times 10^6$ |
| $E(t_0)$ | initial number of exposed individuals | $1.695 \times 10^3$ |
| $I_p(t_0)$ | initial number of post-latent individuals | 308.8 |
| $I_m(t_0)$ | initial number of asymptomatic/mildly symptomatic individuals | 462.4 |
| $I_s(t_0)$ | initial number of severely symptomatic (hospitalized) individuals | 127.4 |
| $Q(t_0)$ | initial number of quarantined individuals | 93.7 |
| $R(t_0)$ | initial number of recovered individuals | 311.1 |
| $M(t_0)$ | initial value of the information index | 101.9 |
| $\Lambda$ | net inflow of susceptibles | $1.762 \times 10^3$ d$^{-1}$ |
| $\mu$ | natural death rate | 10.7/1000 yr$^{-1}$ |
| $\mathcal{R}_0$ | basic reproduction number | 3.49 |
| $\beta_b$ | baseline transmission rate | 2.25 d$^{-1}$ |
| $\beta_0$ | mandatory social distancing transmission rate | $0 - 0.74\beta_b$ |
| $\varepsilon_p$ | modification factor w.r.t. transmission from $I_p$ | 1 |
| $\varepsilon_m$ | modification factor w.r.t. transmission from $I_m$ | 0.033 |
| $\varepsilon_s$ | modification factor w.r.t. transmission from $I_s$ | 0.034 |
| $\rho$ | latency rate | 1/5.25 d$^{-1}$ |
| $\eta$ | post-latency rate | 1/1.25 d$^{-1}$ |
| $p$ | fraction of post-latent individuals developing no/mild symptoms | 0.92 |
| $\gamma_0$ | mandatory quarantine rate | 0.057 d$^{-1}$ |
| $\sigma_m$ | rate at which members of $I_m$ class hospitalize | 0.044 d$^{-1}$ |
| $\sigma_q$ | rate at which quarantined individuals hospitalize | 0.001 d$^{-1}$ |
| $\delta$ | disease-induced death rate | 0.022 d$^{-1}$ |
| $v_m$ | recovery rate for asymptomatic/mildly symptomatic individuals | 0.145 d$^{-1}$ |
| $v_s$ | recovery rate for severely symptomatic (hospitalized) individuals | 0.048 d$^{-1}$ |
| $v_q$ | recovery rate for quarantined individuals | 0.035 d$^{-1}$ |

**Table 2.** Information-dependent parameters values for model (2.1)–(2.2).

| parameter | description | baseline value |
|---|---|---|
| $\alpha$ | reactivity factor of voluntary change in contact patterns | $6 \times 10^{-7}$ |
| $D$ | reactivity factor of voluntary quarantine | $9 \times 10^{-6}$ |
| $\zeta$ | $1 - \zeta$ is the ceiling of overall quarantine rate | 0.01 d$^{-1}$ |
| $a$ | inverse of the average information delay $T_a$ | 1/3 d$^{-1}$ |
| $k$ | information coverage | 0.8 |

collected information, so $T_a = a^{-1}$. We assume that people react in response to information and rumours regarding the daily number of quarantined and hospitalized individuals. The *information coverage k* is assumed to be positive and $k \leq 1$, which mimics the evidence that COVID-19 official data could be under-reported in many cases [8,29].

With this choice, by applying the linear chain trick [30], we obtain the differential equation ruling the dynamics of $M$

$$\dot{M} = a(k(Q + I_s) - M).\tag{2.2}$$

## 2.3. Formulation of the balance equations

Here, we derive in details each balance equation of model (2.1).

### 2.3.1. Equation (2.1a): susceptible individuals, S(t)

Susceptibles are the individuals who are healthy but can contract the disease. The susceptible population increases by the net inflow $\Lambda$, incorporating both new births and immigration (for further details, see electronic supplementary material, §S3), and decreases by natural death—with natural death rate $\mu$—and following infection.

It is believed that COVID-19 is primarily transmitted from symptomatic people (mildly or severely symptomatic). In particular, although severely symptomatic individuals are isolated from the general population by hospitalization, they are still able to infect hospitals and medical personnel [31,32] and, in turn, give rise to transmission from hospital to the community. The pre-symptomatic transmission (i.e. the transmission from infected people before they develop significant symptoms) is also relevant: specific studies revealed an estimate of 44% of secondary cases during the pre-symptomatic stage from index cases [33]. The importance of the asymptomatic transmission (i.e. the contagion from a person infected with COVID-19 who does not develop symptoms) is a controversial matter [34,35]. However, available evidence suggests that asymptomatic individuals are much less likely to transmit the virus [36]. We also assume that quarantined individuals are fully isolated and therefore unable to transmit the disease.

In summary, the compartments of individuals capable to transmit the disease are $I_p$, $I_s$ and $I_m$, which contains not only asymptomatic but also mildly symptomatic individuals.

The routes of transmission from COVID-19 patients as described above are included in the *Force of Infection* (FoI) function, i.e. the *per capita* rate at which susceptibles contract the infection. Quarantine at home during the lockdown led to the substantial separation of quarantined individuals from the general population. For this reason, we consider the quarantine-adjusted FoI [37], given by

$$\text{FoI} = \beta(M) \frac{\varepsilon_p I_p + \varepsilon_m I_m + \varepsilon_s I_s}{N - Q}.$$

The transmission coefficients for $I_p$, $I_m$ and $I_s$ are given by $\varepsilon_p \beta(M)$, $\varepsilon_m \beta(M)$ and $\varepsilon_s \beta(M)$, respectively, with $0 \le \varepsilon_p, \varepsilon_m, \varepsilon_s < 1$.

The function $\beta(M)$, which models how the information affects the transmission rate, is defined as a piecewise continuous, differentiable and decreasing function of the information index $M$, with $\beta(\max(M)) > 0$. We assume that

$$\beta(M) = \pi(c_b - c_0 - c_1(M)),\tag{2.3}$$

where $\pi$ is the probability of getting infected during a person-to-person contact and $c_b$ is the baseline contact rate. In (2.3), we represent the reduction in social contacts through the sum of two social distancing contact rates: the constant rate $c_0$, which represents the choice of social distancing due to the restrictive measures imposed by the government, and an information-dependent voluntary rate $c_1(M)$, with $c_1(\cdot)$ increasing with $M$ and $c_1(0) = 0$. In order to guarantee the positiveness, we assume $c_b > c_0 + \max(c_1(M))$. Following [28], we finally set

$$\beta(M) = \frac{\beta_b - \beta_0}{1 + \alpha M},\tag{2.4}$$

namely $\pi c_b = \beta_b$ (baseline transmission rate), $\pi c_0 = \beta_0$ (mandatory social distancing transmission rate) and $\pi c_1(M) = \alpha M (\beta_b - \beta_0)/(1 + \alpha M)$, where $\alpha$ is a positive constant tuning the reactivity factor of voluntary change in contact patterns. For illustrative purposes, see electronic supplementary material, figure S1A.

### 2.3.2. Equation (2.1b): exposed individuals, E(t)

Exposed (or latent) individuals are COVID-19 infected but are not yet infectious, i.e. capable of transmitting the disease to others. Such individuals arise as the result of new infections of susceptible individuals. The population is diminished by development at the infectious stage (at rate $\rho$) and natural death.

### 2.3.3. Equation (2.1c): post-latent individuals, $I_p(t)$

We assume that after the end of the latency period, the individuals enter a phase where they are infectious and asymptomatic. We call this phase *post-latency* [38] (other authors call it *pre-symptomatic* phase [11] or *prodromic* phase [7]). Post-latent individuals belong to two groups: a *truly asymptomatic* group $p_a I_p$ (people that have no symptoms throughout the course of the disease) and a *pre-symptomatic* group $(1 - p_a)I_p$ (people who develop symptoms at the end of such a phase). The latter, in turn, splits into two subgroups: $p_m(1 - p_a)I_p$ will develop mild symptoms, and $(1 - p_m)(1 - p_a)I_p$ will develop severe symptoms. In our model, we take $p = p_a + p_m(1 - p_a)$. Post-latent individuals diminish due to natural death or because they enter the compartment of asymptomatic/mildly symptomatic individuals $I_m$ (at a rate $p\eta$) or that of severely symptomatic individuals $I_s$ (at a rate $(1 - p)\eta$).

### 2.3.4. Equation (2.1d): asymptomatic/mildly symptomatic individuals, $I_m(t)$

This compartment includes both the asymptomatic individuals, that is infected individuals who do not develop symptoms, and mildly symptomatic individuals [11]. Mildly symptomatic individuals are the only symptomatic individuals that move freely (as far as they can). There is no clear evidence of the relevance of asymptomatic individuals in the COVID-19 transmission. However, asymptomatic individuals test positive in screenings (pharyngeal swabs) and therefore are a part of the count of official diagnoses. Members of this class come from the post-latent stage and get out due to quarantine (at an information-dependent rate $\gamma(M)$), worsening symptoms (at rate $\sigma_m$), recovery (at rate $v_m$) and natural death.

### 2.3.5. Equation (2.1e): severely symptomatic individuals (hospitalized), $I_s(t)$

Severely symptomatic individuals are isolated from the general population by hospitalization. They arise: (i) as consequence of the development of severe symptoms by mild illness (the infectious of the class $I_m$ or the quarantined $Q$); (ii) directly from the fraction $1 - p$ of post-latent individuals that rapidly develop severe illness. This class diminishes by recovery (at rate $v_s$), natural death and disease-induced death (at rate $\delta$).

### 2.3.6. Equation (2.1f): quarantined individuals, $Q(t)$

Quarantined individuals $Q$ are those who are separated from the general population.

The basic idea is to characterize the quarantined compartment in a way that its temporal evolution can be compared with official data. Therefore, we assume that quarantined individuals are asymptomatic/mildly symptomatic individuals. As a matter of fact, the Italian government daily released the number of detected COVID-19 positive cases, which was approximately given by quarantined at home and hospitalized individuals. Self-isolation of susceptible and post-latent individuals is implicitly incorporated in the social distancing term. As for exposed individuals, the permanence in that class is shorter than the infectious classes, hence the potential self-isolation effect of this population on the model dynamics is considered negligible here. We point out that other compartments, like the susceptible or the exposed compartments, could be also split into quarantined and non-quarantined individuals (e.g. [39]).

Quarantined individuals diminish by natural death, aggravation of symptoms (at rate $\sigma_q$, so that they move to $I_s$) and recovery (at a rate $v_q$).

Quarantine may arise in two different ways. On one hand, individuals may be detected by health authorities and daily checked. Such active health surveillance ensures also that the quarantine is, in some extent, respected. On the other hand, a fraction of quarantined individuals chooses self-isolation since they are confident in the government handling of the crisis or just believe the public health messaging and act in accordance [40].

We assume that the final choice to respect or not respect the self-quarantine depends on the awareness about the status of the disease in the community. Therefore, we define the information-dependent quarantine rate as follows:

$$\gamma(M) = \gamma_0 + \gamma_1(M), \tag{2.5}$$

where the rate $\gamma_0$ mimics the fraction of the asymptomatic/mildly symptomatic individuals $I_m$ that has been detected through screening tests and is 'forced' into home isolation. The rate $\gamma_1(M)$ represents the undetected fraction of individuals that adopt quarantine by voluntary choice as result of the influence of the circulating information $M$. The function $\gamma_1(\cdot)$ is required to be a piecewise continuous, differentiable and increasing function w.r.t. $M$, with $\gamma_1(0) = 0$. As in [25,26], we set

$$\gamma_1(M) = (1 - \gamma_0 - \zeta)\frac{DM}{1 + DM}, \tag{2.6}$$

where $D$ is a positive constant tuning the reactivity factor of voluntary quarantine, and $\zeta$ is a constant such that $0 < \zeta < 1 - \gamma_0$. The quantity $1 - \gamma_0 - \zeta$ is the value of the quarantine rate by voluntary choices $\gamma_1(M)$ that can be reached in the case of a high level of circulating information (i.e. a high level of social alarm, ideally represented by $M \to +\infty$). This means that the total quarantine rate $\gamma(M) = \gamma_0 + \gamma_1(M)$ reaches a ceiling value of $1 - \zeta$ under circumstances of very high perceived risk. A representative trend of $\gamma(M)$ is displayed in electronic supplementary material, figure S1B.

### 2.3.7. Equation (2.1g): recovered individuals, $R(t)$

After the infectious period, individuals from the compartments $I_m$, $I_s$ and $Q$ recover at rates $v_m$, $v_s$ and $v_q$, respectively. Natural death is also considered. We assume that recovered individuals acquire long-lasting immunity against COVID-19, although this is a currently debated question (as of 22 May 2020) and there is still no evidence that COVID-19 antibodies protect from re-infection [41].

## 3. The reproduction numbers

A frequently used indicator for measuring the potential spread of an infectious disease in a community is the *basic reproduction number*, $\mathcal{R}_0$, namely the average number of secondary cases produced by one primary infection over the course of the infectious period in a fully susceptible population. If the system incorporates control strategies, then the corresponding quantity is named the *control reproduction number* and is usually denoted by $\mathcal{R}_C$ (obviously, $\mathcal{R}_C < \mathcal{R}_0$).

The reproduction number can be calculated as the spectral radius of the *next generation* matrix $FV^{-1}$, where $F$ and $V$ are defined as Jacobian matrices of the new infection appearance and the other rates of transfer, respectively, calculated for infected compartments at the disease-free equilibrium [42]. In this specific case, if $\beta(M) = \beta_b$ and $\gamma(M) = 0$ in (2.1)–(2.2), namely when containment interventions are not enacted, we obtain the expression for $\mathcal{R}_0$; otherwise, the corresponding $\mathcal{R}_C$ can be computed. Simple algebra yields

$$\mathcal{R}_0 = \beta_b \rho \left[ \frac{\varepsilon_p}{B_1 B_2} + \frac{\varepsilon_m p \eta}{B_1 B_2 B_3} + \frac{\varepsilon_s (1-p) \eta}{B_1 B_2 B_6} + \frac{\varepsilon_s p \eta \sigma_m}{B_1 B_2 B_3 B_6} \right]$$

and

$$\mathcal{R}_C = (\beta_b - \beta_0) \rho \left[ \frac{\varepsilon_p}{B_1 B_2} + \frac{\varepsilon_m p \eta}{B_1 B_2 B_4} + \frac{\varepsilon_s (1-p) \eta}{B_1 B_2 B_6} + \frac{\varepsilon_s p \eta \sigma_m}{B_1 B_2 B_4 B_6} + \frac{\varepsilon_s p \eta \gamma_0 \sigma_q}{B_1 B_2 B_4 B_5 B_6} \right], \tag{3.1}$$

with

$$B_1 = \rho + \mu, \qquad B_2 = \eta + \mu, \qquad B_3 = \sigma_m + v_m + \mu,$$
$$B_4 = \gamma_0 + \sigma_m + v_m + \mu, \quad B_5 = \sigma_q + v_q + \mu, \qquad B_6 = v_s + \delta + \mu.$$

A more detailed derivation and interpretation of the reproduction numbers are given in electronic supplementary material, §S2.

## 4. Parametrization

Numerical simulations are performed in Matlab [43]. We use the `ode45` solver for integrating the system and the platform-integrated functions for getting the plots.

The epidemiological parameters of the model as well as their baseline values are reported in table 1. In the same table, simulation time frame and initial conditions are given. A detailed derivation of such quantities is reported in electronic supplementary material, §S3 and §S4.

In the next subsections, we focus on the numerical implementation of lockdown restrictions and ensuing changes in social behaviour.

### 4.1. The effects of the lockdown on transmission

We explicitly reproduce in our simulations the effects of the restrictions posed to human mobility and human-to-human contacts in Italy. Their detailed sequence is summarized in electronic supplementary material, §S5.

Because data early in an epidemic are inevitably incomplete and inaccurate, our approach has been to try to focus on what we believe to be the essentials in formulating a simple model. Keeping this in mind, we assume that the disease transmission rate occurs in just two step reductions (modelled by the mandatory social distancing transmission rate $\beta_0$ in (2.4)), corresponding to

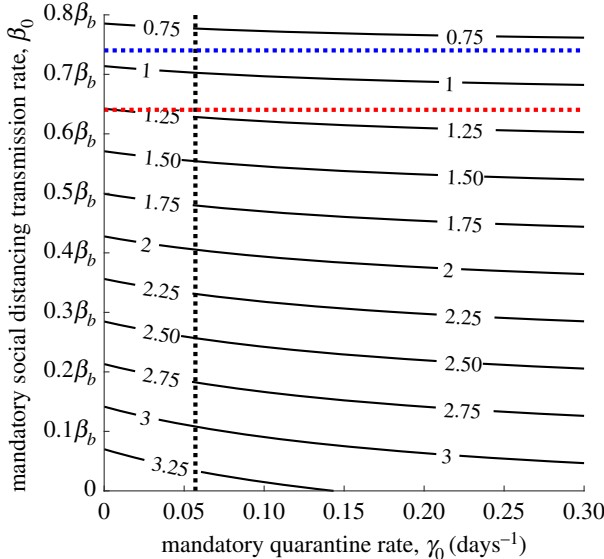

**Figure 2.** Contour plot of the control reproduction number (3.1) versus mandatory quarantine and social distancing transmission rates. Intersection between dotted black and red (resp. blue) lines indicates the value after the first (resp. second) step reduction. Other parameters values are given in table 1.

— 12 March (day 17), i.e. after the first rapid succession of lockdown decrees [5], which cumulatively resulted in a sharp decrease of SARS-CoV-2 transmission;
— 23 March (day 28), that is the starting date of the *full* lockdown [4] that definitely impacted the disease incidence.

In the wake of [8,9], we account for a first step reduction by 64% (that is $\beta_b - \beta_0 \,|_{17 \leq t < 28} = 0.36\beta_b$), which drops the control reproduction number (3.1) close to 1 (see figure 2, dotted black and red lines). It is then strengthened by about an additional 28%, resulting in a global reduction by 74% ($\beta_b - \beta_0 \,|_{t \geq 28} = 0.26\beta_b$) that definitely brings $\mathcal{R}_C$ below 1 (see figure 2, dotted black and blue lines).

## 4.2. Information-dependent parameters

The information-related parameter values are reported in table 2 together with their baseline values.

Following [25,26], we set $\zeta = 0.01$ d$^{-1}$ potentially implying an asymptotic quarantine rate of 0.99 d$^{-1}$ if we could let $M$ go to $+\infty$. As mentioned in §2, the positive constants $\alpha$ and $D$ tune the information-dependent *reactivity*. In particular, $\alpha$ is the reactivity factor of voluntary changes in contact pattern by susceptible and infectious individuals; $D$ is the reactivity factor of voluntary quarantine by individuals with no or mild symptoms. Since the variability of contact rate is strongly affected by limitations imposed by government decrees, we assume that the reactivity in choosing self-isolation in response to information is greater than the reactivity in reducing contacts, that is $D > \alpha$.

The range of values for the information coverage $k$ and the average time delay of information $T_a = a^{-1}$ are mainly assumed or taken from papers where the information index $M$ is used [25,26,28,44]. The information coverage $k$ may be seen as a 'summary' of two opposite phenomena: the disease under-reporting, and the level of media coverage of the status of the disease, which tends to amplify the social alarm. It is assumed to range from a minimum of 0.2 (i.e. the public awareness is 20%) to 1. The average time delay of information $T_a$ ranges from the case of prompt communication (say, $T_a = 1$ day) to the case of large delay (say, $T_a = 60$ days).

The baseline values of the parameters $\alpha$, $D$, $k$ and $a$ are obtained by comparing the model solutions with the official data regarding the number of hospitalized individuals ($I_s$), the number of quarantined individuals ($Q$) and the cumulative deaths as released every day since 24 February 2020 by the Italian Civil Protection Department and archived on GitHub [6].

We get $\alpha = 6 \times 10^{-7}$ and $D = 9 \times 10^{-6}$. With this choice, numerical simulations not displayed here show that the maximum order of magnitude reached by the information index $M$ in the time span considered is

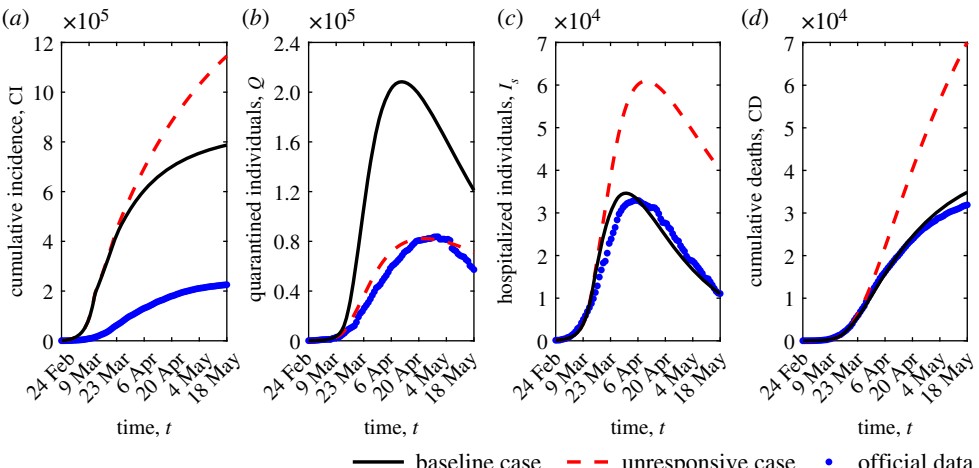

**Figure 3.** Epidemic evolution predicted by model (2.1)–(2.2): cumulative incidence (*a*), quarantined individuals (*b*), hospitalized individuals (*c*) and cumulative deaths (*d*). The predicted evolution (black solid lines) is compared with the unresponsive case $\alpha = D = 0$ (red dashed lines) and with official data (blue dots). Parameter values are given in tables 1 and 2.

equal to $10^5$. Moreover, we obtain $k = 0.8$ and $T_a = 3$ days, meaning a level of awareness about the daily number of quarantined and hospitalized individuals of 80%, resulting from the balance between underestimates and media amplification and inevitably affected by rumours and misinformation spreading on the web (the so-called infodemic [45]). Such awareness is not immediate, but information takes on average 3 days to be publicly disseminated, the communication being slowed by a series of articulated procedures: timing for swab tests results, notification of cases, reporting delays between surveillance and public health authorities, and so on.

Of course, parameters setting is influenced by the choice of curves to fit. Available data seem to provide an idea about the number of identified infectious people who have developed mild/moderate symptoms (the fraction that mandatorily stays in $Q$) or more severe symptoms (the hospitalized, $I_s$) and the number of deaths, but much less about those asymptomatic or with very mild symptoms who are not always subjected to a screening test.

# 5. Numerical results

Let us consider the time frame $[t_0, t]$, where $t_0 \leq t \leq t_f$. We consider two relevant quantities, the *cumulative incidence* CI($t$), i.e. the total number of new cases in $[t_0, t]$, and the *cumulative deaths* CD($t$), i.e. the disease-induced deaths in $[t_0, t]$.

For model (2.1)–(2.2), we have, respectively

$$\text{CI}(t) = \int_{t_0}^{t} \beta(M(\tau)) \frac{S(\tau)}{N(\tau) - Q(\tau)} (\varepsilon_p I_p(\tau) + \varepsilon_m I_m(\tau) + \varepsilon_s I_s(\tau)) \, d\tau,$$

where $\beta(M)$ is given in (2.4), and

$$\text{CD}(t) = \int_{t_0}^{t} \delta I_s(\tau) \, d\tau.$$

In figure 3, the time evolution in $[t_0, t_f]$ of CI($t$) and CD($t$) is shown (figure 3*a* and figure 3*d*), along with that of quarantined individuals $Q(t)$ (figure 3*b*) and hospitalized individuals $I_s(t)$ (figure 3*c*). The role played by information on the public compliance with mitigation measures is stressed by the comparison of the baseline scenario with the *unresponsive* case ($\alpha = D = 0$ in (2.1)–(2.2)), that is the case when circulating information does not affect disease dynamics. Corresponding dynamics are labelled by black solid and red dashed lines, respectively.

In the unresponsive case, the cumulative incidence is much less impacted by the lockdown restrictions in comparison with the baseline scenario ($11.45 \times 10^5$ versus $7.85 \times 10^5$ on 18 May). Furthermore, in this case, the quarantined individuals given by the model are only those that choose self-isolation when 'forced' by public health authorities after detection. That is to say, the quarantined

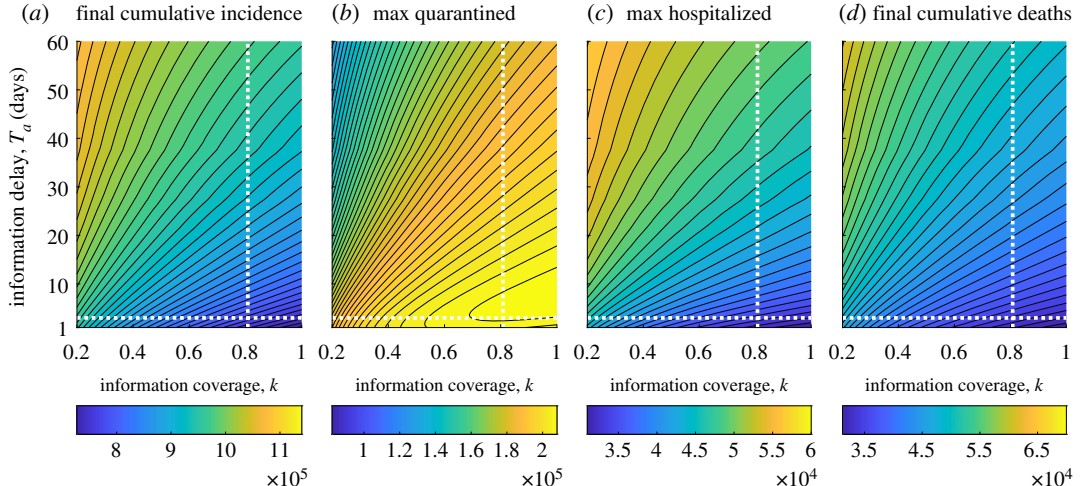

**Figure 4.** Contour plots of relevant quantities versus information coverage $k$ and average delay $T_a = a^{-1}$. (a) Cumulative incidence $CI(t_f)$ evaluated at the last day of the considered time frame, i.e. $t_f = 85$, corresponding to 18 May 2020. (b) Peak of quarantined individuals $\max(Q)$. (c) Peak of hospitalized individuals $\max(I_s)$. (d) Final cumulative deaths $CD(t_f)$. The intersection between dotted white lines indicates the values corresponding to the baseline scenario $k = 0.8$, $T_a = 3$ days. Other parameter values are given in tables 1 and 2.

individuals predicted by the model reduce to those ones officially detected (i.e. what is counted in the official data). As a consequence, the *peak* of hospitalized patients is about 77% higher and 10 days time-delayed, with a corresponding increase in cumulative death of more than 100%. For all reported dynamics, the deviation between the baseline and the unresponsive case starts to be clearly distinguishable after the first step reduction of 64% in transmission rate (on 12 March).

Trends are also compared with officially disseminated data [6] (figure 3, blue dots), which seem to conform accordingly for most of the time horizon, except for CI, which suffers from an inevitable and probably high underestimation [8,9,11,29]. As of 18 May 2020, we estimate about 785 000 infections, whereas the official count of confirmed infections is 225 886 [6].

We now investigate how the information parameters $k$ and $a$ may affect the epidemic course. More precisely, we assess how changing these parameters affects some relevant quantities: the *peak* of quarantined individuals $\max(Q)$ (i.e. the maximum value reached by the quarantined curve in $[t_0, t_f]$), the peak of hospitalized individuals $\max(I_s)$, the cumulative incidence $CI(t_f)$ evaluated at the last day of the considered time frame, i.e. $t_f = 85$ (corresponding to 18 May 2020), and the final cumulative deaths $CD(t_f)$.

The results are shown in the contour plots in figure 4. As expected, $CI(t_f)$, $\max(I_s)$ and $CD(t_f)$ decrease proportionally to the information coverage $k$ and inversely to the information delay $T_a$: they reach the minimum for $k = 1$ and $T_a = 1$ day. Differently, the quantity $\max(Q)$ may not monotonically depend on $k$ and $T_a$ as it happens for $k \geq 0.6$ and $T_a \leq 15$ days (see figure 4b, lower right corner). In such parameter region, for a given value of $k$ (resp. $a$) there are two different values of $a$ (resp. $k$) which correspond to the same value of $\max(Q)$. The absolute maximum ($\max_{[k,T_a]}(\max(Q))$) is obtained for $k = 1$ and $T_a \approx 7$ days. Note that the pair of values $k = 1$, $T_a = 1$ day corresponds to the less severe outbreak, but not with the highest peak of quarantined individuals.

In what follows, we compare the relative changes for these quantities w.r.t. the unresponsive case. In other words, we introduce the index

$$RX = \frac{X - X^0}{X^0},$$

which measures the percentage *relative* change of $X \in \{CI(t_f), \max(Q), \max(I_s), CD(t_f)\}$ w.r.t. the corresponding quantity $X^0$ predicted by model (2.1)–(2.2) with $\alpha = D = 0$.

All the possible values arising in the parameter ranges $k \in [0.2, 1]$ and $T_a \in [1, 60]$ days are shown in electronic supplementary material, figure S2. However, we report in table 3 the results corresponding to the unresponsive case and to three example responsive cases, the baseline and two extremal ones

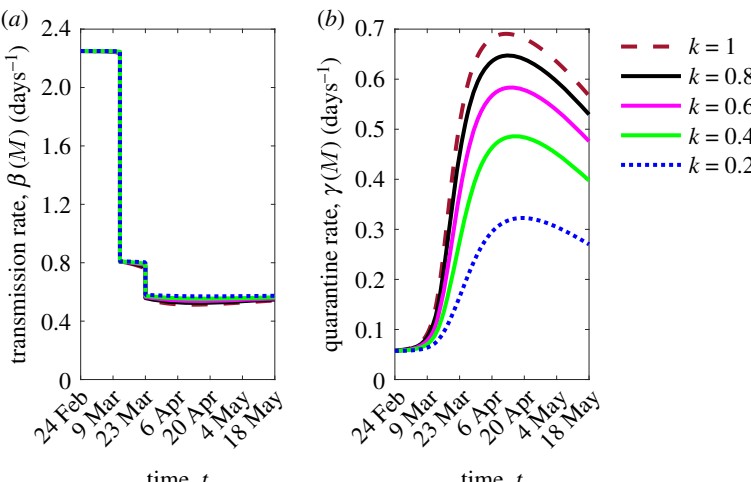

**Figure 5.** Time evolution of the transmission rate (*a*) and the quarantine rate (*b*) of model (2.1)–(2.2), by fixing $T_a = 3$ days. Colour meaning is specified in the figure legend and refers to five values of the information coverage *k*. Other parameter values are given in tables 1 and 2.

**Table 3.** Exact and relative values of final cumulative incidence $CI(t_f)$, the peak of quarantined individuals $\max(Q)$, the peak of hospitalized individuals $\max(I_s)$ and final cumulative deaths $CD(t_f)$, for three combinations of information parameters *k* and $T_a$ (second to fourth row), in comparison with the unresponsive case: $\alpha = D = 0$ in (2.1)–(2.2) (first row). Other parameters values are given in tables 1 and 2.

| case | $CI(t_f)$ | $RCI(t_f)$ | $\max(Q)$ | $Rmax(Q)$ | $\max(I_s)$ | $Rmax(I_s)$ | $CD(t_f)$ | $RCD(t_f)$ |
|---|---|---|---|---|---|---|---|---|
| $\alpha = D = 0$ | $11.45 \times 10^5$ | 0 | $0.82 \times 10^5$ | 0 | $6.09 \times 10^4$ | 0 | $7.00 \times 10^4$ | 0 |
| $k = 0.8$, $T_a = 3$ d | $7.85 \times 10^5$ | −0.31 | $2.08 \times 10^5$ | 1.53 | $3.45 \times 10^4$ | −0.43 | $3.48 \times 10^4$ | −0.50 |
| $k = 1$, $T_a = 1$ d | $7.27 \times 10^5$ | −0.37 | $2.05 \times 10^5$ | 1.49 | $3.10 \times 10^4$ | -0.49 | $3.12 \times 10^4$ | −0.55 |
| $k = 0.2$, $T_a = 60$ d | $10.83 \times 10^5$ | −0.05 | $1.30 \times 10^5$ | 0.58 | $5.70 \times 10^4$ | −0.06 | $6.21 \times 10^4$ | −0.11 |

(i) the baseline scenario $k = 0.8$, $T_a = 3$ days, which is close to the best possible fitting with official data, given the model and the considered parameter ranges;

(ii) the case of highest information coverage and lowest information delay, $k = 1$, $T_a = 1$ day;

(iii) the case of lowest information coverage and highest information delay, $k = 0.2$, $T_a = 60$ days.

Compared with the baseline scenario, a more accurate and faster communication (case (ii)) would drive to a significant reduction of $CI(t_f)$, $\max(I_s)$ and $CD(t_f)$ (more precisely, by 37%, 49% and 55%, respectively, see table 3, third row). Moreover, even the worst possible information-based scenario (case (iii)) is significantly better than the unresponsive case (compare first and fourth rows in table 3).

As mentioned above, information and rumours regarding the status of the disease in the community affect the transmission rate $\beta(M)$ (as given in (2.4)) and the quarantine rate $\gamma(M)$ (as given in (2.5)).

In our last simulation, we want to emphasize the role of the information coverage on the quarantine and transmission rates. In figure 5, a comparison with the case of low information coverage, $k = 0.2$, is given assuming a fixed information delay $T_a = 3$ days (blue dotted lines). It can be seen that more informed people react and quarantine: an increasing of the maximum quarantine rate from 0.32 to 0.69 d$^{-1}$ (which is also reached a week earlier) can be observed when increasing the value of *k* to $k = 1$ (figure 5*b*).

The effect of social behavioural changes is less evident in the transmission rate where increasing the information coverage produces a slight reduction of the transmission rate mainly during the full lockdown phase (figure 5*a*). This reflects the circumstance that the citizens choice of social distancing is not enhanced by the information-induced behavioural changes during the first stages of the epidemic.

# 6. Conclusion

In this work, we propose a mathematical approach to investigate the effects on the COVID-19 epidemic of social behavioural changes in response to lockdowns.

Starting from an SEIR-like model, we assume that the transmission and quarantine rates are partially determined on a voluntary basis and depend on the circulating information and rumours about the disease, modelled by a suitable time-dependent *information index*. We focus on the case of the COVID-19 epidemic in Italy and explicitly incorporate the progressively stricter restrictions enacted by the Italian government, by considering two step reductions in the contact rate (the partial and full lockdowns).

The main results are as follows:

— we estimate two fundamental information-related parameters: the information coverage regarding the daily number of quarantined and hospitalized individuals (i.e. the parameter $k$) and the information delay (the quantity $T_a = a^{-1}$). The estimate is performed by comparing the model's solutions with official data. We find $k = 0.8$, which means that the public was aware of 80% of real data and $T_a = 3$ days, the time lag that was necessary for information to reach the public;

— social behavioural changes in response to lockdowns played a decisive role in curbing the epidemic curve: the combined action of voluntary compliance with social distance and quarantine resulted in preventing a duplication of deaths and about 46% more infections (i.e. approx. 360 000 more infections and 35 000 more deaths compared with the *unresponsive* case, as of 18 May 2020);

— even under circumstances of low information coverage and high information delay ($k = 0.2$, $T_a = 60$ days), there would have been a beneficial impact of social behavioural response on disease containment: as of 18 May, cumulative incidence would be reduced by about 5% and deaths by about 11%.

Shaping the complex interaction between circulating information, human behaviour and epidemic disease is challenging. In this manuscript, we give a contribution in this direction. We provide an application of the information index to a specific field-case, the COVID-19 epidemic in Italy, where the information-dependent model is parametrized and the solutions are compared with official data.

Our study presents limitations that leave the possibility of future developments. In particular: (i) the model captures the epidemics at a country level but it does not account for regional or local differences and for internal human mobility (the latter having been crucial in Italy at the early stage of the COVID-19 epidemic). (ii) The model does not explicitly account for ICU admissions. The limited number of ICU beds constituted a main issue during the COVID-19 pandemics [46]. This study does not focus on this aspect but ICU admissions could be certainly included in the model. (iii) The model could be extended to include age structure. Age was particularly relevant for COVID-19 lethality rate (in Italy, the lethality rate for people aged 80 or over was more than double the average value for the whole population [47]).

Further developments may also concern the investigation of optimal intervention strategies during the COVID-19 epidemics and, to this regard, the assessment of the impact of vaccine arrival. In this case, the approach of information-dependent vaccination could be employed [24,26,44].

Data accessibility. Numerical simulations are performed in Matlab version R2019b by routinely implemented algorithms. We use the ode45 solver for integrating the model system and the platform-integrated functions for getting the plots. §4 in the article provides all the necessary information to reproduce the results. The work makes use of official data of COVID-19 epidemic in Italy as released and stored on GitHub by the Italian Civil Protection Department [6].

Authors' contributions. B.B. conceived and coordinated the study. B.B. and R.D.M. designed the model. R.D.M. parametrized the model and performed the numerical experiments. B.B. and R.D.M. wrote and revised the manuscript.

Competing interests. We declare we have no competing interest.

Funding. We received no funding for this study.

Acknowledgements. The authors are grateful to the anonymous reviewers for their constructive comments, which have improved the earlier version of the manuscript. The present work has been performed under the auspices of the Italian National Group for the Mathematical Physics (GNFM) of National Institute for Advanced Mathematics (INdAM).

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
